# Painting with light-powered bacteria

Jochen Arlt [1], Vincent A. Martinez [1], Angela Dawson[1], Teuta Pilizota [2] & Wilson C.K. Poon[1]

Self-assembly is a promising route for micro- and nano-fabrication with potential to revolutionise many areas of technology, including personalised medicine. Here we demonstrate that external control of the swimming speed of microswimmers can be used to self assemble reconfigurable designer structures in situ. We implement such 'smart templated active self assembly' in a fluid environment by using spatially patterned light fields to control photon-powered strains of motile *Escherichia coli* bacteria. The physics and biology governing the sharpness and formation speed of patterns is investigated using a bespoke strain designed to respond quickly to changes in light intensity. Our protocol provides a distinct paradigm for self-assembly of structures on the 10 µm to mm scale.

[1] School of Physics and Astronomy, The University of Edinburgh, Peter Guthrie Tait Road, Edinburgh EH9 3FD, UK. [2] School of Biological Sciences and Centre for Synthetic and Systems Biology, The University of Edinburgh, Alexander Crum Brown Road, Edinburgh EH9 3FF, UK. Correspondence and requests for materials should be addressed to J.A. (email: j.arlt@ed.ac.uk)

There are two conceptually distinct ways to construct nano- and micro-structures: 'top-down' techniques such as lithography use 'scalpels' (chemical etching or electron beams, etc.), whereas microscopic 'Lego components' move themselves into position—self assemble—in 'bottom-up' techniques[1]. Both equilibrium phase transitions (e.g., crystallisation) and non-equilibrium processes are exploited for self assembly. In either case, external templates can be used to direct the process, with reconfigurable templates offering programmability[2]. Self-assembly was originally inspired by viral biology, where the components are biomacromolecules[3]. Increasingly, synthetic colloidal building blocks are used, with bespoke particle shape, size and interaction, e.g., 'patchy particles' with heterogeneous surface chemistry[4]. Active, or self-propelled, colloids open up further opportunities[5, 6].

Such self-propelled micro-swimmers are attracting significant recent attention[7] as 'active matter'[8, 9]. They violate time-reversal symmetry[10] and may be used to transport colloidal 'cargos'[11, 12]. For both fundamental physics and applications, external control of swimming, e.g., using particles with light-activated self-propulsion[13–16], opens up many new possibilities. For example, light-activated motile bacteria[17] can be used to actuate and control micro-machinery[18].

The self-assembly of micro-swimmers into clusters of tens of particles has already been demonstrated[16, 19, 20]. Recent simulations[21] suggest that the patterned illumination of light-activated swimmers can be used for the templated self assembly[1] of designer structures comprising $10^3$–$10^4$ particles. Real-time reconfiguration of the light field would then allow smart templated active self assembly (STASA), which we here implement for the first time using light-controlled motile bacteria[17].

An *E. coli* bacterium swims[22] by turning $\approx 7$–$10\,\mu$m long helical flagella attached to the cell body ($\approx 2\,\mu$m $\times 1\,\mu$m) using membrane-embedded rotary motors powered by a protonmotive force (PMF), which arises from the active pumping of $H^+$ to the extracellular medium[23]. Unlike all synthetic active colloids to date and most bacteria, *E. coli* can generate PMF in nutrient-free motility buffer[24] by utilising internal resources and oxygen ($O_2$) to produce energetic electron pairs. These release their energy stepwise along an electrochemical potential ladder of respiratory enzymes located in the inner cell membrane, generating a PMF of $\approx -150$ mV. The electron pair ultimately passes to and reduces $O_2$ to water. Thus, with no $O_2$, PMF = 0 and swimming ceases[22]. If cells under anaerobic conditions can express proteorhodopsin (PR)[17], a green-photon-driven proton pump[25], then they will swim only when suitably illuminated: these are living analogues of synthetic light-activated microswimmers[13–16].

In the following we demonstrate STASA using light-controlled *E. coli* bacteria. We show that the rapidity with which such bacteria react to changes in light intensity has a major influence on the pattern formation, and we construct a bespoke strain for which this response is fast on the 1 s time scale. We characterise the dynamics of pattern formation and explain the different scaling laws observed for protocols in which the bacteria are either initially stationary or already swimming. Finally, we explore the physics controlling the sharpness and resolution of patterns.

## Results

### Temporal response of light-controlled *E. coli* strains.
We suspended cells in phosphate motility buffer at optical density OD $\lesssim$ 8 (cell-body volume fraction $\approx 1.1\%$)[22] and sealed 2 µl into 20 µm-high, flat capillaries, where cells swim in two dimensions but have enough room to 'overtake' each other in all three spatial dimensions. Differential dynamic microscopy (DDM)[26] returned an averaged speed $\bar{v} \approx 30\,\mu$m s$^{-1}$ and a non-motile fraction $\beta \approx 20\%$ at OD = 1 under fully oxygenated conditions. (We use 'non-motile' to refer to cells that can never swim and 'stationary' for non-swimming cells capable of motility when illuminated.)

Motile cells were allowed to swim until $O_2$ was depleted and $\bar{v}$ dropped abruptly to zero after a few minutes[22] (see Supplementary Fig. 1). After these cells were left in the dark for $\approx 10$ min, green illumination was turned on (510–560 nm, intensity $\mathcal{I} \approx 5$ mW cm$^{-2}$ at the sample). The stationary cells accelerated uniformly before saturating in speed (Fig. 1a). Fitting the data for freshly prepared cells to $\bar{v}(t) = \bar{v}_{sat} t/(t + \tau_{on})$ gives $\bar{v}_{sat} = 9.5$ µm s$^{-1}$, $\tau_{on} = 30$ s. $\bar{v}_{sat}$ can be controlled by the illumination intensity $\mathcal{I}$ (Fig. 1a, inset), up to $\lesssim 28\,\mu$m s$^{-1}$. We find that $\{\bar{v}, \beta, v_{sat}, \tau_{on}\}$ changed over hours as cells aged.

What happens when illumination ceases can be optimised by genetic modification. In the strain we constructed for this work (AD10), in which the *unc* gene cluster has been deleted (see Methods), $\bar{v}$ has dropped sharply by the time the first data point was taken after the illumination was switched off, so that $\tau_{off} < 1$ s, the sampling period in this part of the experiment (Fig. 1b). It is unclear why a few cells (< 1%) continued to swim, so that $\bar{v}$ never quite reached zero. In a strain without *unc* deletion (AD57), $\bar{v}$ dropped much slower upon cessation of illumination (Fig. 1b) as seen previously[17, 27]. The *unc* cluster codes for $F_1F_o$-ATPase, which is normally powered by the PMF to generate ATP. However, in darkness the *unc* wild type can use this membrane protein complex in reverse to export protons and so sustains a PMF for some time[28]; this is not possible with *unc* deletion[29].

In the absence of any source of PMF in darkness, $\tau_{off}$ should be controlled by the discharging of the PMF through an effective resistor–capacitor circuit (see Supplementary Note 1 for details). The effective membrane capacitance is $C \gtrsim 10^{-14}$ F. The effective resistance, $R_{tot}$, combines in parallel the resistance of the membrane, the motors (each of which is $R_{mot} \lesssim 10^{14}\,\Omega$) and other transmembrane components, so that $R_{tot} < R_{mot}$ is an upper bound, and we expect $R_{tot}C < 1$ s, as observed. Consistent with this interpretation, $\tau_{off}$ is approximately independent of the starting speed of decelerating cells at our experimental time resolution (Fig. 1c).

The fitted 'on time' for freshly prepared cells, $\tau_{on} \approx 30$ s (Fig. 1a, black points), is likely controlled by the rate constant[30] for stator units to come on and off motors, $k_{stator} \approx 0.04$s$^{-1} \sim \tau_{on}^{-1}$. In sustained darkness, motors disassemble in PR-bearing *E. coli*, and full 'motor resurrection' upon illumination takes $\sim 200$ s [27], in agreement with our data (Fig. 1a). Consistent with this, if illumination is lowered and then restored rapidly (i.e., within a time interval $\ll k_{stator}^{-1}$), motors do not have time to disassemble and $\tau_{on} \approx \tau_{off}$ (Fig. 1d).

### Smart templated active self assembly.
Our STASA protocol uses a digital mirror device to project patterns of illumination onto a uniform field of cells rendered stationary by $O_2$ exhaustion. We first projected a positive mask with bright features on a dark background (Fig. 2a, inset). The width of the features, 90 µm, is comparable to the persistence length of the smooth-swimming *E. coli* strains used in this work (i.e., the length scale over which cells effectively swims in straight lines)[31]. Over time, the mask pattern was replicated in the cell population with increasing clarity: cells swim out of illuminated areas, stop within $\tau_{off} \sim 1$ s and accumulate at the edges just outside the boundary of these areas (Fig. 2b and see also Supplementary Movie 1). Next, we abruptly switched to a negative mask with the same pattern (dark pattern against bright background; Fig. 2c, inset). Edge-accumulated cells start swimming, about half of which arrive inside the dark

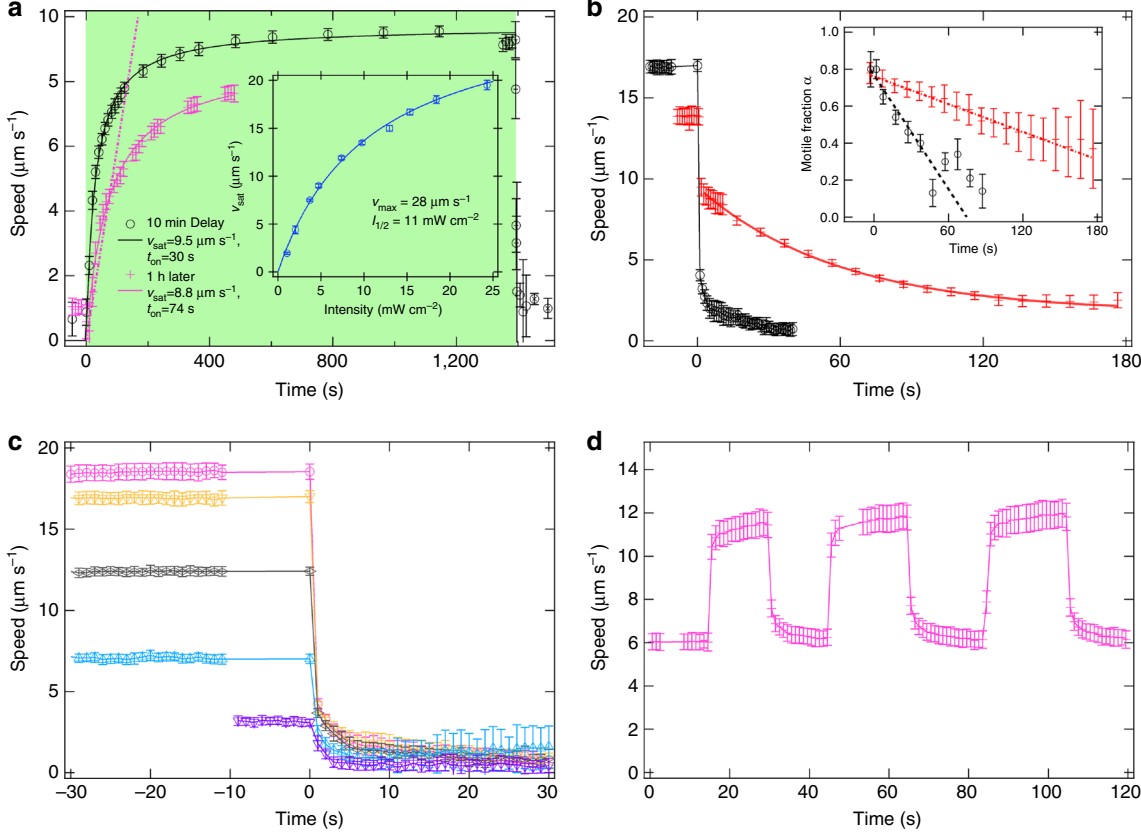

**Fig. 1** Response to light. **a** Population-averaged speed vs. time for initially stationary *E. coli* AD10, light on at $t = 0$ and off at $t \approx 1{,}400$ s (black circles); green shading = illuminated, and the data point just after switch-off are 1 s apart. This sample was measured soon after the cells had run out of $O_2$. One hour later, cells accelerated less rapidly (magenta crosses). Full lines: fits to $\bar{v}(t) = \bar{v}_{sat} t / (t + \tau_{on})$. The dot-dashed line shows the acceleration ($0.05\ \mu\text{m s}^{-2}$) obtained from 'box emptying' experiments starting from stationary cells. Inset: dependence of $\bar{v}_{sat}$ on incident light intensity (points) fitted to $\bar{v}_{sat}(\mathcal{I}) = \bar{v}_{max}\mathcal{I}/(\mathcal{I} + \mathcal{I}_{1/2})$ (see Supplementary Note 1). **b** Strain AD10 with *unc*-cluster deletion shows a rapid decrease in $\bar{v}$ when illumination ceases (black circles). In a strain without *unc*-cluster deletion (AD57), a significant fraction of the population (see inset for the motile fraction $\alpha = 1 - \beta$, where lines are guides to the eye) continues to swim at reduced speed, resulting in a much slower decay of $\bar{v}$ (red crosses; solid red line: offset exponential fit, giving $\tau \approx$ 60 s). **c** Average swimming speed of AD10 as green light is switched off (at $t = 0$): $\tau_{off} \approx 1$ s independent of the initial swimming speed (at the resolution limit of our technique). **d** AD10 bacteria adapt their swimming speed within 1 s as the green light is alternated between low and high intensity. In this case $\tau_{on} \approx \tau_{off}$. Error bars in all these plots depict SD from the mean speed over a range of $q = 0.5$–$1.5\ \mu\text{m}^{-1}$

features immediately and stop within $\tau_{off}$, rapidly outlining the inside edge of the masked areas (Fig. 2c). Thereafter, the dark areas progressively filled up: bacteria continually arrive from the outside, which acts as an essentially infinite reservoir of swimming cells (Fig. 2d, e and see also Supplementary Movies 2 and 3). Other patterns can be 'painted' using this method (Fig. 2g), their shape can be updated in situ and the patterns can be erased by illuminating them with a uniform light field (see Supplementary Movie 4). In contrast to the recently demonstrated painting of one-off polychromatic patterns using mixed colonies of differently coloured bacteria on solid agar[32], our method produces dynamic monochrome patterns that are reconfigurable in real time inside a liquid medium.

**Dynamics of pattern formation**. To understand how fast spatial patterns can be established, we experimented with square masks (Fig. 2f) starting from two initial conditions: a (dark) field of stationary cells ($\bar{v}(0) = 0$, i.e., diffusive) and a field of (illuminated) cells that have reached speed saturation ($\bar{v}(0) = \bar{v}_{sat}$). We studied the evolution of cell density (see Methods) and the speed inside bright squares against a dark background. Starting with diffusive cells, $\bar{v}(t) = \bar{v}_{sat} t / (t + \tau_{on}) \approx a t$ with $a = \bar{v}_{sat}/\tau_{on}$ for

$t \lesssim \tau_{on}$. The time to empty an $L \times L$ square should thus scale as

$$t_D(L) \sim (2L/a)^{0.5}. \qquad (1)$$

The measured evolution of the density of 1 h-old cells, $\rho_{tot}(t)$, in illuminated squares of different sizes is shown in Fig. 3a, from which we extracted $t_D$, the time for $\rho_{tot}$ to drop to $1/e$ of its decay. As expected, $t_D(L) \sim L^{0.5}$ (Fig. 3c). Fitting Eq. (1) gives $a = 0.05$ $\mu\text{m s}^{-2}$, which is a reasonable average acceleration over the first $\sim$ 100 s for the magenta data points in Fig. 1 for 1 h-old cells (where the dot-dash line has slope $0.05\ \mu\text{m s}^{-2}$). Moreover, $\bar{v}_{D,max}$, the maximum average swimming speed reached during emptying depends on the box size, displaying a $L^{0.5}$ scaling, which is also consistent with our model (Fig. 3c and Supplementary Note 2).

On the other hand, starting with cells all swimming (on average) at $\bar{v}_{sat}$, a square should empty in

$$t_S(L) \sim L/\bar{v}_{sat}. \qquad (2)$$

From the measured $\rho_{tot}(t)$ in this case (Fig. 3b), we deduced a 'box emptying time' and find (Fig. 3c) the expected scaling, $t_S(L) \sim L$. Fitting Eq. (2) gives $\bar{v}_{sat} \approx 8\ \mu\text{m s}^{-1}$, consistent with the measured $\bar{v}_{sat} = 8.8\ \mu\text{m s}^{-1}$. Equations (1) and (2) are also

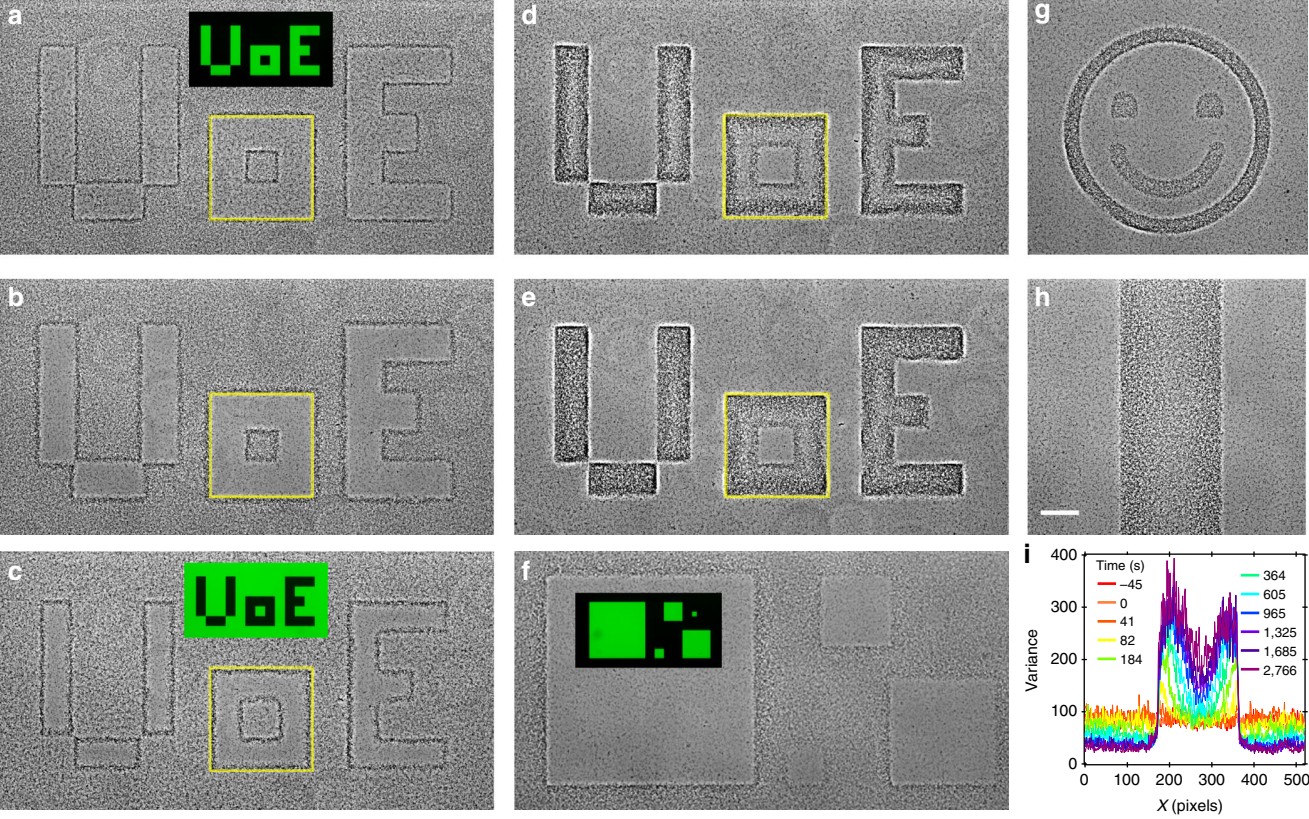

**Fig. 2** Smart templated active self-assembly. Positive and negative masks projected onto initially stationary cells: **a** sample illuminated for 1 min with a positive 'UoE' (University of Edinburgh) pattern shown in inset, **b** same after 9 min of illumination, **c** 1 min after switching to the negative pattern shown in inset (with green = bright), as well as **d** 6 min and **e** 12 min after switching. Yellow square = boundary between light and dark regions of the 'o'. **f** Samples illuminated with pattern of squares shown in inset. **g** Negative smiley pattern. **h** Negative strip together with **i** time evolution of the strip's density profile. Scale bar in **h** is 100 μm and applies throughout

confirmed by the data collapse achieved for $\rho_{tot}(t)$ when we scale the time axis by $L^{0.5}$ and $L$, respectively (Fig. 3a, b).

Further insight on the time scale for feature formation comes from studying cells moving into a dark area. We projected a 270 μm-wide dark strip onto a field of initially diffusive cells (Fig. 2h). An edge rapidly developed just inside the darkened area, corresponding to twin peaks in the spatial cell density profile (Fig. 2i; as measured by the variance of image intensity, see Methods), which broaden and fill the stripe over time. The squared displacement of this peak shows diffusive dynamics, $\Delta x^2 = 2Dt$ (Fig. 3d) with a fitted $D = 0.74\,\mu\text{m}^2\,\text{s}^{-1}$. The interface between bright and dark areas acts as an absorber of swimmers arriving with an effective diffusive process. Owing to conservation of mass, the diffusive propagation of the peak is thus expected, although quantitative aspects remain to be elucidated.

**Pattern sharpness and resolution.** Finally, we seek to understand what controls the sharpness of the self-assembled features using our method. A stark demonstration of the answer comes from projecting a step-function pattern of light–dark illumination on initially uniformly swimming cells (see Supplementary Note 3 for details). Figure 4 shows the resulting cell density profile (from the intensity variance, see Methods) for two strains, AD10, with $\tau_{stop} \lesssim 1\,\text{s}$, and AD57, a strain lacking the *unc* cluster deletion and having $\tau_{off} > 1\,\text{min}$. In the case of AD10, the illumination pattern is clearly replicated in cell density within 1 min and the steady state shows a sharp feature with half width $\lesssim 10\,\mu\text{m}$. For AD57, it takes $\gg 1\,\text{min}$ to establish a corresponding density pattern, which, in the steady state, is $\gtrsim 50\,\mu\text{m}$ wide. Thus, a short enough

$\tau_{off}$, or more specifically a short mean stopping distance $\bar{l}_{stop} = \bar{v}\tau_{off}$, is necessary for the fast establishment of sharp features using our protocol.

The resolution achievable with STASA depends on the mean stopping distance $\bar{l}_{stop}$ of the bacteria, as well as the sharpness and contrast of the projected light template. In our current experimental implementation, the optics projecting the templating pattern is designed to illuminate a large area of the sample at the expense of spatial resolution. This allowed us to reliably construct features of about 10 μm in width (Fig. 5). However, ultimately the stopping distance $\bar{l}_{stop}$ will prove the practical limit of STASA, which at low swimming speeds is comparable to the size of individual bacteria. The use of more complex grayscale templates and dynamically adjustable patterns will help reaching this limit more rapidly and reliably.

## Discussion

We have demonstrated how to perform smart templated self assembly using a proteorhodopsin-bearing strain of *E. coli* with *unc* cluster deletion dispersed in anaerobic motility buffer. Spatial patterns of light are reproduced sharply as patterns in cell density. The time scale of pattern formation is controlled by the (saturation) swimming speed and the 'on time', and the sharpness of the features by the 'off time'. Once established, the patterns persist for as long as the light template is applied. Uniform light can be used to quickly erase the patterning, whereas removing the light completely will lead to a slow diffusion driven 'dissolution' of the pattern.

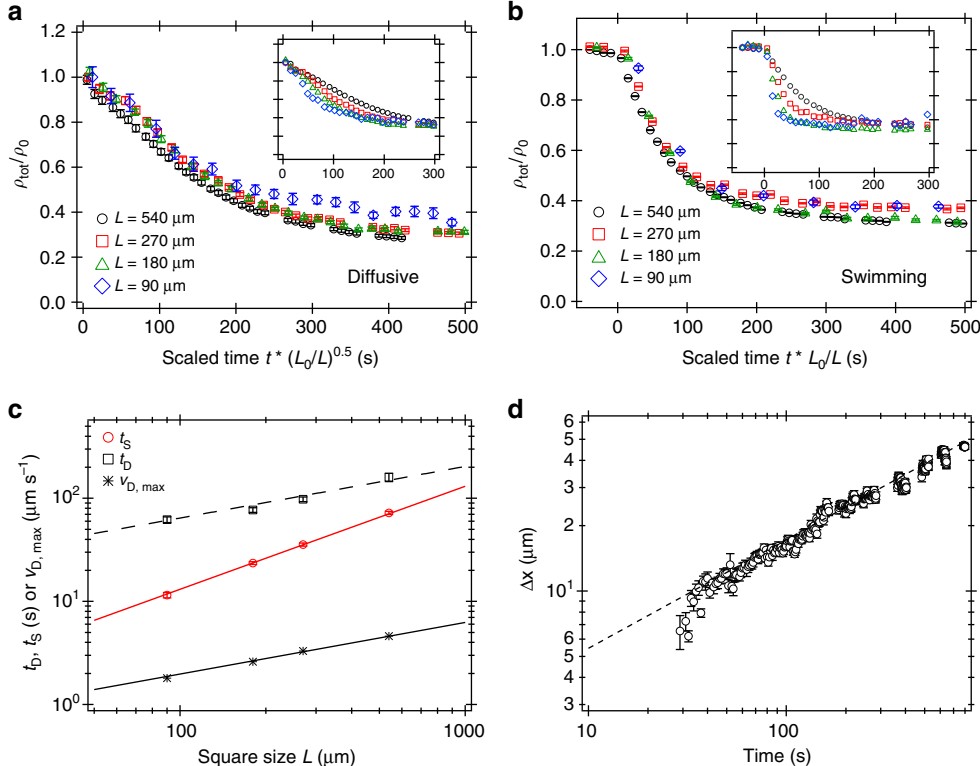

**Fig. 3** Characterising the dynamics of pattern formation. Time evolution of normalised bacterial density (as measured by DDM, see Methods) inside squares of different dimensions $L$ (inset, Fig. 2f), starting from **a** diffusive state and **b** uniformly swimming state. Inset: data plotted against time; main plots: data collapse when scaling time by $\sqrt{L_0/L}$ in **a** and $L_0/L$ in **b** (both with $L_0 = 540\ \mu m$). **c** Scaling of 'box emptying times' $t_D$, $t_S$, as well as $\bar{v}_{D,max}$ (the maximum swimming speed reached when starting with diffusive cells) with square size $L$. **d** Shift of the peak position over time in the dark stripe (Fig. 2h); dashed line = power law fit of slope 0.5. Error bars in **a**, **b** are SE of the $q$-averages and in **c**, **d** the SD estimates returned by the fitting routines

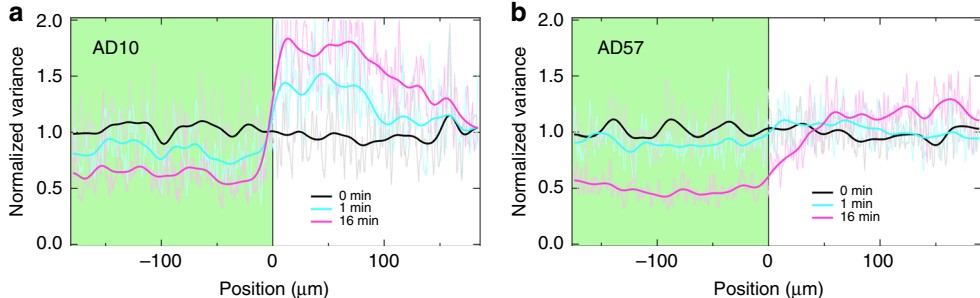

**Fig. 4** Effect of stopping distance. Response to a step illumination pattern (left bright, right dark) projected at time $t = 0$ by two strains starting from initially uniformly swimming cells. Faint lines show the normalised variance (a measure of cell density) profile, whereas bold lines show numerically smoothed profiles. **a** AD10 with $\tau_{off} \lesssim 1\,s$ and **b** AD57 with $\tau_{off} \gtrsim 1\,min$ (corresponding images in Supplementary Fig. 3)

The successful implementation of this protocol opens up many opportunities. In terms of fundamental active matter physics, it should now be possible to test the fundamental prediction that the product of density and speed is constant in a system with spatially dependent activity[33]. For practical applications, it should now be possible to demonstrate an 'all biological' version[21] of the controlled actuation of lithographed micro-devices using light-activated bacteria[18]: STASA allows the assembly and real-time reconfiguration of the devices themselves using the same bacteria used to actuate the devices. On the other hand, if high cell density is used to trigger the secretion of adhesive biopolymers, and/or bacteria can be turned into microparticles of refractory ceramic-precursor materials such as silica[34], then bacterial STASA offers a route for manufacturing bespoke 10–100 μm microstructures[35],

which remains a challenging length scale for current additive manufacturing methods.

## Methods

**Bacterial strains**. Both AD10 and AD57 are smooth-swimming mutants, which were constructed using our laboratory stock of *E.coli* AB1157[36]. For AD10, all of the genes present within the operon encoding the ATP synthase complex (*atp*I, *atp*B, *atp*E, *atp*F, *atp*H, *atp*A, *atp*G, *atp*D, *atp*C) were deleted in a single step by allelic replacement using a recombinant pTOF24 vector[37] containing 400 bp homology arms flanking the deletion on each side. This resulted in the Δ*unc* strain AD10 with a deletion of 7504 bp (corresponding to position 3,915,552–3,923,056 of the *E.coli* K12 MG1655 chromosome). P1 transduction was used to delete the *che*Y gene in both this strain and the parental strain (to generate AD57) using JW1871 (BW25113 *che*Y) as donor[38]. Diagnostic PCR reactions were performed to confirm genotypes. Both strains were transformed with plasmid pBAD-His C (a kind gift

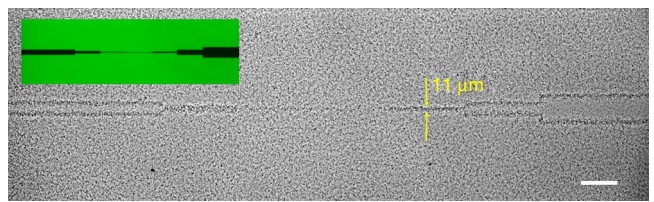

**Fig. 5** Spatial resolution for AD10 strain. A pattern of dark lines of different width (see inset; nominal line width on the sample from left to right: 45, 22, 6, 3, 11, 45 and 90 μm) is projected onto a dense (OD ≈ 8) sample of AD10. At low swimming speeds ($v = 2.7\ \mu m\ s^{-1}$) the 11 μm-wide line still shows up clearly while the density contrast for the thinner lines is very low. Scale bar: 100 μm

from Jan Liphardt, UC Berkley) encoding the SAR86 γ-proteobacterial photo-rhodopsin variant[27].

**Sample preparation**. Overnight cultures were grown aerobically in 10 ml Luria–Bertani broth using an orbital shaker at 30 °C and 200 r.p.m. A fresh culture was inoculated as 1 : 100 dilution of overnight-grown cells in 35 ml tryptone broth and grown for 4 h, by which point AD10 has reached an optical density of $OD_{600} \approx$ 0.2 (whereas AD57 reached OD ≈ 0.35). The production of PR was induced by adding arabinose to a concentration of 1 mM, as well as the necessary cofactor all-*trans*-retinal to 10 μM in the growth medium. Cells were incubated under the same conditions for a further hour to allow protein expression to take place and then transferred to motility buffer (MB, pH 7.0, 6.2 mM $K_2HPO_4$, 3.8 mM $KH_2PO_4$, 67 mM NaCl and 0.1 mM EDTA). Cells were re-suspended in MB after a single filtration (0.45 μm HATF filter, Millipore), yielding high-concentration stock solutions (OD ≈ 8–10), which were then diluted with MB as required.

Suspensions were loaded into 2 μl sample chambers (SC-20-01-08-B, Leja, NL, ≈ 6 mm × 10 mm × 20 μm), giving a quasi-two-dimensional environment for *E. coli* swimming. The chamber was sealed using vaseline to stop flow and oxygen supply, and placed onto a microscope for video recording. *E. coli* initially swims under endogenous metabolism, which depletes the oxygen within the sample cells[22]. Once oxygen is depleted, the cells stop swimming unless illuminated by green light.

**Optical setup**. The samples were observed using a Nikon TE2000 inverted microscope with a phase-contrast objective (PF 10×/0.3). Time series of ≈ 40 s movies were recorded at 100 frames per second using a fast CMOS camera (MC 1362, Mikrotron). A long-pass filter (RG630, Schott Glass) was introduced into the bright-field light path to ensure that the light required for imaging did not activate PR. The light controlling the swimming behaviour of the bacteria was provided by an LED light source (Sola SE II, Lumencor) filtered to a green wavelength range (510–560 nm) to overlap with the absorption peak of PR[17]. This light was reflected off a digital mirror device (DLP6500, Texas Instruments), which was imaged onto the microscope sample plane using demagnifying relay optics (see Supplementary Fig. 4 for a schematic of the complete optical setup). The light was coupled into the transmission path by introducing a dichroic mirror in between the long working distance phase-contrast condenser and the sample. This 'trans' geometry allowed us to uniformly illuminate a circular area of ≈ 2.9 mm in diameter, larger than the objective's field of view. As the digital mirror chip is conjugate with the sample plane, any pattern applied to it is directly imaged onto the sample, with the edge length of a digital mirror 'pixel' corresponding to 2.7 μm. Computer control of the pattern displayed on the digital mirror device gave us precise spatial and temporal control of the illumination pattern. For the work presented here, we restricted ourselves to binary patterns and simply toggled between a few selected patterns, although the setup is capable of projecting dynamic grayscale patterns.

**Motility and density measurements**. The motility of the samples was quantified using DDM, a fast high-throughput technique for measuring the dynamics of particle suspensions such as colloids[39] or swimming microorganisms[22, 26, 40]. For swimming bacteria, DDM measures motility parameters averaged over ∼ $10^4$ cells from ≈ 30 s-long movies via the differential image correlation function $g(\vec{q}, \tau)$, i.e., the power spectrum of the difference between pairs of images delayed by time $\tau$. Under appropriate imaging conditions and for isotropic motion, $g(\vec{q}, \tau)$ is related to the intermediate scattering function $f(q, \tau)$, which is the $q$th Fourier component of the density temporal autocorrelation function, via

$$g(q, \tau) = A(q)[1 - f(q, \tau)] + B(q). \tag{3}$$

Here, $B(q)$ relates to the background noise and $A(q)$ is the signal amplitude. Fitting $f(q, \tau)$ to a suitable swimming model of *E. coli*, we obtain the average, $\bar{v}$, and width, $\sigma$, of the speed distribution $P(v)$, the fraction of non-motile bacteria $\beta$ and their diffusion coefficient $D$[40]. Provided the system is dilute enough such that its structure factor $S(q) \approx 1$, $A(q)$ is proportional to the particle density[41, 42]. Thus, in

two otherwise identical samples with different cell densities $\rho_{1,2}$,

$$\frac{A_1(q)}{A_2(q)} = \frac{\rho_1}{\rho_2}. \tag{4}$$

We verified this using bacteria suspensions of different cell concentrations (measured by optical density), Supplementary Fig. 5.

However, the spatial resolution of this method of density measurement is very limited. In order to quantify the dynamics of the edge formation we needed to extract density profiles (Fig. 2i) from images such as the band shown in Fig. 2h. We found that neither the intensity along a line nor the average of such line profiles bears much resemblance to the cell density variation visible to direct inspection (see Supplementary Fig. 6). This is related to the nonlinearities at high phase shifts and artifacts inherent to phase contrast imaging[43].

Interestingly, the single line profile (Supplementary Fig. 6b) shows that the intensity fluctuates more in the high-density region. Indeed, the averaged horizontal variance profile (Supplementary Fig. 6d) shows qualitative agreement with the visually observed density profile of the band, suggesting that the variance (with background subtracted) could be used to quantify local density. Supplementary Fig. 5 shows that, indeed, densities of uniform samples deduced from variance measurements agree quantitatively with densities deduced from DDM using Eq. (4) and from optical density measurements. At least within the low density limit, this scaling of intensity variance with cell density is not surprising: a (dark) cell on a uniform (light) background will lower the mean intensity slightly and introduce a characteristic amount of variance relative to this background. As more cells are added, they initially increase the variance by the same characteristic amount (as the mean background hardly changes), leading to the observed proportionality. For phase-contrast imaging, a phase-dark object is often surrounded by a lighter halo, so that the net effect of cells on mean intensity is rather small while at the same time leading to a significant characteristic variance per object.

**Data availability**. The research data presented in this publication is available on the Edinburgh DataShare repository[44].

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

## Acknowledgements
The work is funded by an EPSRC programme grant (EP/J007404/1) and a ERC Advanced Grant (ADG-PHYSAPS). We thank M.E. Cates for discussions of active self assembly and J. Walter for discussions on strain design.

## Author contributions
J.A., V.A.M., W.C.K.P. and T.P. designed the research. A.D. designed, constructed and verified all the strains. J.A. and V.A.M. performed the experiments. J.A., V.A.M. and W.C.K.P. analysed the data. All authors contributed to the interpretation and discussion of results, and W.C.K.P., J.A. and V.A.M. wrote the manuscript.

## Additional information

**Competing interests:** The authors declare no competing financial interests.

