## [Peer Review File · Nature Communications]

Reviewers' comments:

Reviewer #1 (Remarks to the Author):

The authors use *E. coli*, which are genetically engineered to move in the presence of light, to reproduce light patterns generated with a DMD. The reported effect works because the bacteria motility is suppressed in the absence of the light stimulus (i.e. *E. coli* only show diffusive behavior), while it reactivates in its presence (i.e. *E. coli* show enhanced diffusivity). Ultimately, when a light pattern is imposed, this induces a spatial variation in the bacterial motility that translates into the reproduction of the virtual boundaries imposed with the light template.

I certainly find the main result of the paper interesting and worth reporting in *Nature Communications* after revision. In particular, I invite the authors to expand/discuss/clarify the following points:

- The authors describe their results as self-assembly throughout the manuscript. I would rather consider this as a nice example of templating or directed assembly of active matter rather than self-assembly as there is a significant external influence on the pattern formation.
- Is the literature really exhaustive when it comes to active particles' response to boundaries imposed by external fields or templating of colloids with optical patterns? For example, although the physical mechanisms described are different, these recent articles show conceptually relevant results: *Nature Communications* 4, 2588 (2013); *Nature Communications* 7, 10694 (2016); *Nature Communications* 7, 10907 (2016). Similarly, when it comes to templating colloids with optical fields, it might be worth referring to, e.g., *Nature Communications* 5, 3676 (2014).
- I found the description of the light patterns generation with the DMD rather short and definitely not enough to reproduce the experiments independently. What experimental configuration are the authors using? Are they modulating intensity or is it a phase-only modulation? What are the masks given to the DMD? How is the imaging of the chip done on the sample? These are important details that should be provided and detailed either in the main text or in the supplementary.
- One of the main advantages of the authors design compared to previous literature (e.g. *Nature Chemical Biology* 13, 706 (2017)) is the fact that they can reconfigure the bacteria among different patterns using the DMD. However, this important experimental scenario has not been considered or characterized in detail.
- What is the spatial resolution of this method, i.e. what is the smallest feature that can be produced reliably? What are the aspects that limit the resolution in the system?
- Similarly, in Fig.2i the contrast of the pattern seems to reduce in time. Have the authors characterized the pattern stability beyond the times reported in Fig.2i? Would the patterns slowly disappear even under the stimulus of the light pattern?

- The patterns in Fig.2 are mainly obtained by projecting a “positive image” followed by a “negative image”. What would happen if this order is inverted?

Some other minor points to consider:

- In the abstract, I would say that the main conceptual distinction to make is between “top-down” and “bottom-up” techniques of which lithography and self-assembly are two examples.

- I found the analogy of bacteria as “active living colloids” a bit stretched and confusing in this paper’s context. Although artificial and biological microswimmers share analogies, I wonder whether in this case this analogy is really necessary rather than a digression.

- On page 3, define beta immediately or point to the Methods section.

- In Fig.1, it could be useful to indicate when the light is on/off directly on the plot for better clarity.

Reviewer #2 (Remarks to the Author):

This manuscript describes a process where a 2D microscale pattern arises in the form of spatial bacterial cell density variation controlled by an external light module. To achieve this, the authors have engineered an E.coli strain that can start/stop moving when the illumination light is switched on/off. The ability of achieving a sharp contrast pattern is enabled by the unique property of this engineered bacterial strain -- fast stop time when the light is switched off. While a proteorhodopsin bacterial mutant that responds to light exists in the literature, the authors further deleted an unc gene cluster such that the response time to dark is shortened.

The ability to generate microscale patterns using light module as a control has the potential to provide a platform for studies of mechanisms of how nano-motors work, physics of collective dynamics, as well as for engineering new class of smart materials. In this sense, the manuscript is timely. The manuscript is clearly written, and the message is of interests to general public.

The manuscript has a clear and detailed characterization of the temporal motility response of the bacteria to light, and the temporal process of pattern formation when illuminated with light. However, to make the claim that this is a programmable self assembly process, one needs to say more about the spatial resolution of this process. What is the smallest feature that can be made by this method? Is there an upper limit? What is the fidelity from the mask to the actual fabricated

pattern. This can be achieved by a systematic study of pattern formation with features of different sizes, or a theoretical computation and discussion. A minor concern is on the presentation. The key step that makes this possible is the fast response time of the bacteria, which is made possible by unc gene cluster deletion. The impact of unc cluster deletion should be emphasized in the main text (e.g. by including it in the abstract, by adding time response data of AD57 to Fig.1).

NCOMMS-17-25849-T

Response to reviewers' comments:

Reviewer #1 (Remarks to the Author):

The authors use *E. coli*, which are genetically engineered to move in the presence of light, to reproduce light patterns generated with a DMD. The reported effect works because the bacteria motility is suppressed in the absence of the light stimulus (i.e. *E. coli* only show diffusive behavior), while it reactivates in its presence (i.e. *E. coli* show enhanced diffusivity). Ultimately, when a light pattern is imposed, this induces a spatial variation in the bacterial motility that translates into the reproduction of the virtual boundaries imposed with the light template.

I certainly find the main result of the paper interesting and worth reporting in *Nature Communications* after revision. In particular, I invite the authors to expand/discuss/clarify the following points:

- The authors describe their results as self-assembly throughout the manuscript. I would rather consider this as a nice example of templating or directed assembly of active matter rather than self-assembly as there is a significant external influence on the pattern formation.

The referee has a valid point in emphasising the crucial role of the light pattern, which evidently provides the essential external influence on the pattern formation. However, we note that the swimming orientation of bacteria is not directed by the light pattern, i.e. individual bacteria swim in random directions. Thus we believe that referring to the process as (smart) **templated** (active) self assembly [STASA] in the title and throughout the manuscript is appropriate. Additionally, the work discussed in the manuscript conforms to the 'definition' proposed in our reference [1] as 'processes that involve pre-existing components, are reversible and can be controlled by proper design of the components'. Furthermore, closely related work such as our ref. 21 [Stenhammar *et al.* *Sci. Adv.* 2 (2016)] also use the term 'self-assembly'. We would therefore prefer to keep the term as it is.

- Is the literature really exhaustive when it comes to active particles' response to boundaries imposed by external fields or templating of colloids with optical patterns? For example, although the physical mechanisms described are different, these recent articles show conceptually relevant results: *Nature Communications* 4, 2588 (2013); *Nature Communications* 7, 10694 (2016); *Nature Communications* 7, 10907 (2016). Similarly, when it comes to templating colloids with optical fields, it might be worth referring to, e.g., *Nature Communications* 5, 3676 (2014).

We thank the referee for pointing out these additional references, which indeed all help to set the context for our work. We have added the suggested references to the manuscript.

- I found the description of the light patterns generation with the DMD rather short and definitely not enough to reproduce the experiments independently. What experimental configuration are the authors using? Are they modulating intensity or is it a phase-only modulation? What are the masks given to the DMD? How is the imaging of the chip done on the sample? These are important details that should be provided and detailed either in the main text or in the supplementary.

We have extended the description of the optical setup to make it more complete. In particular, we now state explicitly that the DMD chip is imaged onto the sample plane and have also added a schematic of the setup as a supplementary figure.

- One of the main advantages of the authors design compared to previous literature (e.g. Nature Chemical Biology 13, 706 (2017)) is the fact that they can reconfigure the bacteria among different patterns using the DMD. However, this important experimental scenario has not been considered or characterized in detail.

We agree with referee that fast reconfigurability is a very important aspect of our work. We now provide evidence of fast dynamic reconfiguration with the 'UoE' pattern in figure 2b-2c; in particular SI movie 2 shows the pattern being inverted in real time.

To demonstrate further such reconfigurability we have added a further supplementary movie, which shows the transformation of the smiley pattern shown in fig 2g into a sad face, which is then erased completely; we have updated the main text accordingly.

We agree that characterising the dynamics of reconfiguration is important. However, since our method can be used to generate a variety of patterns, it is impractical to characterise the formation dynamics in many individual cases. Instead, we have reported in some detail experiments on the time scales involved in forming various simple patterns (squares of different sizes). The scaling relationships obtained should enable the reader to estimate the relevant length and time scales in their own experiments using any desired pattern up to numerical prefactors of order unity.

- What is the spatial resolution of this method, i.e. what is the smallest feature that can be produced reliably? What are the aspects that limit the resolution in the system?

In principle, the smallest feature depends on the stopping distance, which is a function of the swimming speed and the stopping time. In practice, the sharpness and modulation contrast of the projected mask also play a role. We have added a new section to the manuscript discussing this explicitly. We also include some new experimental data showing that the resolution for our current system is ~10 microns.

- Similarly, in Fig.2i the contrast of the pattern seems to reduce in time. Have the authors characterized the pattern stability beyond the times reported in Fig.2i? Would the patterns slowly disappear even under the stimulus of the light pattern?

It is not quite clear to us what the referee means by 'the pattern seems to reduce in time'. The density contrast between the dark band and the illuminated area actually keeps increasing throughout the time series. However, the contrast between the centre of the dark band and its inside edges decrease as the feature fills up, slowly approaching a more uniform density throughout the band. The diffusive dynamics of this process is discussed in the main text. To make it easier to check these features visually, we provide a preview of the time-lapse videos of the band formation (which are part of the dataset supporting the manuscript) on the following link (Password: Referee): <https://datasync.ed.ac.uk/index.php/s/DNW9uWNYLi9IoIS>.

While the sample is illuminated the pattern is stable and slowly approaches a dynamic steady state that reproduces the shape of the template more and more faithfully, e.g. by assuming a more uniform density in the dark areas. The new time-lapse video extends for 55 min, by which point very little change can be observed.

The patterns 'dissolves' only fairly slowly after the illumination is switched off. The process is driven by diffusion, which is relatively slow, particularly (as in our case) when it occurs next to a surface. The proximity of a surface and the high density of bacteria reduce the diffusion constant well below the free-particle value of $D = 0.3 \mu\text{m}^2/\text{s}$.

We have clarified these points in the revised manuscript.

- The patterns in Fig.2 are mainly obtained by projecting a "positive image" followed by a "negative image". What would happen if this order is inverted?

As a first approximation the order of projecting positive and negative masks does not matter. However, "negative" patterns tend to concentrate bacteria in the dark areas, leading to very high cell densities. Inverting the pattern once such high concentrations have been achieved does lead to new phenomena such as collective motion and instabilities of the boundaries. We are in the process of analysing these collective phenomena in detail, which will form the subject of a future publication.

Some other minor points to consider:

- In the abstract, I would say that the main conceptual distinction to make is between "top-down" and "bottom-up" techniques of which lithography and self-assembly are two examples.

We thank the referee for this suggestion as it indeed makes the conceptual distinction between the two approaches more explicit. We have updated our introductory paragraph.

- I found the analogy of bacteria as "active living colloids" a bit stretched and confusing in this paper's context. Although artificial and biological microswimmers share analogies, I wonder whether in this case this analogy is really necessary rather than a digression.

This analogy is certainly not necessary in the context of this work, and although it is an established expression within the active matter field we have decided to remove it from the manuscript.

- On page 3, define beta immediately or point to the Methods section.

We now define β explicitly as the non-motile fraction.

- In Fig.1, it could be useful to indicate when the light is on/off directly on the plot for better clarity.

We have added green shading to Fig. 1 to indicate the time period during which the light was on.

Reviewer #2 (Remarks to the Author):

This manuscript describes a process where a 2D microscale pattern arises in the form of spatial bacterial cell density variation controlled by an external light module. To achieve this, the authors have engineered an E.coli strain that can start/stop moving when the illumination light is switched on/off. The ability of achieving a sharp contrast pattern is enabled by the unique property of this

engineered bacterial strain -- fast stop time when the light is switched off. While a proteorhodopsin bacterial mutant that responds to light exists in the literature, the authors further deleted an *unc* gene cluster such that the response time to dark is shortened.

The ability to generate microscale patterns using light module as a control has the potential to provide a platform for studies of mechanisms of how nano-motors work, physics of collective dynamics, as well as for engineering new class of smart materials. In this sense, the manuscript is timely. The manuscript is clearly written, and the message is of interests to general public.

The manuscript has a clear and detailed characterization of the temporal motility response of the bacteria to light, and the temporal process of pattern formation when illuminated with light.

However, to make the claim that this is a programmable self assembly process, one needs to say more about the spatial resolution of this process. What is the smallest feature that can be made by this method? Is there an upper limit? What is the fidelity from the mask to the actual fabricated pattern. This can be achieved by a systematic study of pattern formation with features of different sizes, or a theoretical computation and discussion.

In principle, the smallest feature depends on the stopping distance, which is a function of the swimming speed and the stopping time. In practice, the sharpness and modulation contrast of the projected mask also play a role. We have added a new section to the manuscript discussing this explicitly. We also include some new experimental data showing that the resolution for our current system is ~10 microns. Moreover, following the suggestion of this reviewer, we now include a new Fig. 5 showing how well different feature sizes can be projected.

There is no clear upper limit to the feature size as it is mostly limited by practical considerations such as the size of the illuminated area (currently around 2.9 mm in diameter) and the resolution of the spatial light modulator.

A minor concern is on the presentation. The key step that makes this possible is the fast response time of the bacteria, which is made possible by *unc* gene cluster deletion. The impact of *unc* cluster deletion should be emphasized in the main text (e.g. by including it in the abstract, by adding time response data of AD57 to Fig.1).

We thank the referee for pointing out this oversight in our presentation. The revised manuscript now indeed emphasizes the impact of *unc* deletion by mentioning it in the abstract, extending figure 1 and moving most of the characterization from the supplementary information to the main manuscript.

REVIEWERS' COMMENTS:

Reviewer #1 (Remarks to the Author):

The authors have made convincing extra experiments to address the points raised by both reviewers. I believe the paper can now be accepted for publication.

Reviewer #2 (Remarks to the Author):

The authors have adequately addressed my concern. I recommend the manuscript to be published in Nature Communication.